# In-Process Cutting Temperature Monitoring Method Based on Impedance Model of Dielectric Coating Layer at Tool-Chip Interface

Heebum Chun [ID], William Park, Jungsub Kim and ChaBum Lee *[ID]





J. Mike Walker 66' Department of Mechanical Engineering, Texas A&M University, 3123 TAMU,
College Station, TX 77843, USA
*    Correspondence: cblee@tamu.edu; Tel.: +1-979-458-8121

**Abstract:** This paper introduces a novel approach to in-process monitoring of the cutting temperature at the tool-chip interface (TCI). Currently, there are no tools available in the commercial market for measuring and monitoring cutting processes at the TCI region. Therefore, most of the studies about evaluating cutting temperature rely on simulation results without knowing the true temperature at the actual TCI region. In addition, recent cutting temperature measurement techniques have measurement errors occurring resulting from difficulty in estimations at the TCI region. However, the proposed method enables the measuring of cutting temperature by directly probing the localized TCI using a cutting tool coated with dielectric material. The study was conducted by utilizing the impedance characteristics of the dielectric outer layer of the cutting tool. A chemical vapor deposition (CVD) diamond coated insert that is commercially available was considered for the study to avoid wear effect. Impedance response of the dielectric layer under varying temperature conditions is assessed by Nyquist diagram using an impedance analyzer. The result of the Nyquist diagram showed temperature-dependent impedance characteristics that showed good agreement with the results from the thermal experiment which was a comparison between impedance response and elevated temperature. The impedance at the TCI for monitoring cutting temperature is measured under a turning process on a lathe using a constant current source. The impedance responses showed a significant decrease in impedance under various machining conditions which indicates a rise in cutting temperature. Moreover, different machining conditions showed different temperature profiles. The impedance responses were further characterized for depth of contact, which found that a drop in impedance corresponded to an increase in depth of contact. Therefore, the study showed that in-process monitoring of the cutting temperature is possible using an impedance model of the dielectric coating layer at the local TCI. Furthermore, with its versatility, this method is expected to measure the vibration, chatters, cutting force, and so on, as the results showed that impedance is not only sensitive to temperature but also to contact area. The application and expectation of this study is to provide real-time machining data to help end users in manufacturing industry to improve product quality, productivity, and prolonged lifespan of cutting tools.

**Keywords:** cutting temperature; in-process monitoring; impedance spectroscopy; dielectric layer; tool-chip interface

## 1. Introduction

A considerable amount of heat is generated during the metal cutting process, due to the shearing and deforming of the workpiece. The heat generated from the machine tool during the cutting process may increase adhesive, abrasive, and diffusive wear on the cutting tool, reducing the tool lifespan. In addition, the cutting temperature increases the chance of getting an edge built up on the tool tip area. For high-performance machining, cutting temperature should be adequately monitored since it is closely related with cutting

force, vibration, and tool wear [1–4]. The final quality of a workpiece varies according to the cutting mechanism as mentioned; therefore, assessing and evaluating cutting temperature are non-negligible [5–7], and an elevated temperature during the machining process should be precisely controlled and monitored since cutting temperature is an important indicator of the machining performance [8–10]. Although monitoring the cutting temperature is crucial, it is difficult to control and monitor as there are mechanical interferences due to physical inaccessibility. As of right now, various methods have been introduced to measure cutting temperatures (i.e., tool-workpiece thermocouple, embedded thermocouple, and tool-insert thermocouple [11–14]). However, these measurement methods cannot measure the true cutting temperature at the local TCI so-called 'hot spot' when cutting is in process. Instead, they measure the distributed or mean temperature of the workpiece and cutting tool by placing thermocouples at nearby locations, and the junction size of the thermocouples affect the dynamic thermal response [15–18]. Moreover, thermocouples may be physically embedded into the workpiece which interferes not only with temperature measurements but also damages the workpiece. Therefore, the cutting temperature measurement methods mentioned above may result in significant errors in its measurements due to the characteristics and usage requirements of the thermocouple.

Non-contact measurement methods include infrared photography, pyrometers, etc., but obvious limitations exist [19,20]. The photographic technique often experiences limited pixel size and resolution. In addition, it is not suitable as an in-process real time monitoring method as it requires a high-speed frame rate which increases the cost of implementation. Exposure time needs to meet adequate specification because average temperature prevails during the exposure period [20]. Pyrometers are incompatible with measuring temperature due to the chip covering the device during the machining process. Such measurement techniques are further affected by thermal background radiation and materials due to different emissivity causing uncertainties and errors in the measurement [21–23]. To overcome such difficulties Lin et al., studied an inverse finite element method with measured surface using an infrared pyrometer to estimate the TCI cutting temperature [24]; and Storchak et al. studied analytical models based on stresses for predicting thermal properties along with software algorithm simulation that includes measured or analytically estimated cutting temperature [25]. In summary, out of all the problems with the non-contact measurement methods, given their inability to measure the true temperature of the cutting region interface, the measurements are merely estimations of the interface region. It is nearly impossible to directly measure the interface temperature of the machining process due to known inaccessibility issues.

To improve the limitations of the methods mentioned above, this study proposes a novel approach to monitor the interface cutting temperature while machining is in process. The monitoring model of in-process machining is based on the impedance characteristics of the cutting tool with a dielectric coating layer. Dielectric coating on on the surface of quality cutting tools is up to tens of microns thick and prolongs the lifespan of the cutting tool by preventing wear, diffusion, oxidation, and thermal shock. The cutting tools are coated with dielectric coating materials such as diamond, ceramics, and cubic boron nitride (CBN) to enhance the ability of the cutting tool, and broaden workpiece material choices and cutting parameters [26,27]. The dielectric coating materials used in cutting tools have high impedance characteristics and are therefore often used in high-output batteries and capacitors [28]. This study utilizes the dielectric impedance formed by the cutting tool in contact with a workpiece during the machining process. In specific terms, the change in relative permittivity of the dielectric material varies with temperature which is the main concept for this proposed method. Hence, in-process monitoring of a cutting process is feasible by analyzing the impedance responses of the dielectric layer. In other words, the cutting process at the local TCI region may be analyzed in real time with the proposed cutting temperature monitoring method based on the impedance response.

## 2. Methodology

During the metal cutting process, the dielectric coating layer of the cutting tool is sandwiched in between the base material of the cutting tool and workpiece as schematically illustrated in Figure 1. The base material of the cutting tool and workpiece conduct electricity, functioning as electrodes, and the dielectric coating layer behaves as a dielectric capacitor. Therefore, the model of the equivalent electric circuit can be constructed with leakage resistance, $R_L$, and parallel capacitance, C. The impedance of the dielectric coating layer is dependent upon its thickness, the contact area between the two electrodes, and the relative permittivity of the dielectric coating layers. The relative permittivity is contingent on temperature which enables measuring the true cutting temperature at the TCI region. The proposed monitoring of the in-process cutting is undertaken once the cutting process begins, and as soon as the cutting tool touches the workpiece, it forms a dielectric capacitor. Thus, the characterization of the electrical impedance response—with its model from the dielectric coating layers—enables in situ monitoring of the interface cutting temperature. The method requires no physical hardware, meaning the cutting tool itself is a cutting temperature measurement sensor. This has a huge advantage as it requires no hardware implementation, thus minimizing uncertainty arising from sensor installations. The measurement process is not hampered by covered chips or inaccessibility of the sensor, as the sensor is directly probing the interface region.

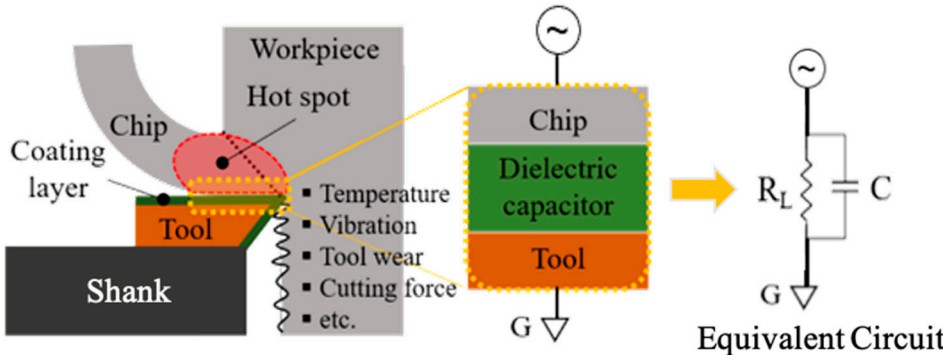

**Figure 1.** Schematic of metal cutting process and corresponding equivalent circuit.

## 3. Validation

*Impedance Spectroscopy*

As previously mentioned, impedance of dielectric material is subject to change under temperature variation due to relative permittivity. In many cases, the electrical properties of the metals show a highly non-linear relationship with temperature change [29,30]. As the temperature increases, electrons from the valence band are able to jump to the conduction band, creating free movement between the two bands, and the metal conductivity changes. Thus, for characterizing the impedance response under given temperature conditions, the impedance analyzer (CHI-604D) was used to examine the impedance behavior under varying temperature throughout the Nyquist diagram. The commercially available CVD diamond coated insert tool manufactured by Stellram with a thickness of 10 μm was used for the study. The diamond coated insert was selected for the entire study to avoid cutting tool wear effect. The parameters of the temperature include room temperature (RT), and 50 °C to 300 °C with 50 °C increments, giving a total of seven conditions. Temperature was measured with a thermocouple attached near to where the chip was forming. A temperature controller was used for the cartridge heater to elevate the temperature at the TCI and for the feedback-control to maintain the given temperature conditions using an attached thermocouple. Nyquist diagrams under varying degrees were obtained while the desired temperature was sustained. The range of the Nyquist diagrams spans from 0.1 to 100 kHz due to the limitation of the impedance analyzer. From Figure 2, Nyquist diagrams showed no parasitic, contact resistance; inductance at room temperature within the range of the diagram; and delineated a 1st order system response. As the temperature

increased from 50 to 300 °C with 50 °C increments, the leakage resistance $R_L$, which is estimated at 390 MΩ at the RT, significantly dropped to 160, 38, 4.5, 0.46, 0.12, and 0.10 MΩ. Furthermore, 45-degree phase shifts with at least two orders at the elevated temperature in the phase diagram are observed, which predict the natural frequency of the dielectric coating material. Due to the limited range of the frequency sweep of the impedance analyzer, the parasitic or contact resistance at higher temperatures could not be observed. Overall, it can be concluded that the impedance is tremendously sensitive and closely related to temperature, and therefore has the ability to measure the cutting temperature during the machining process.

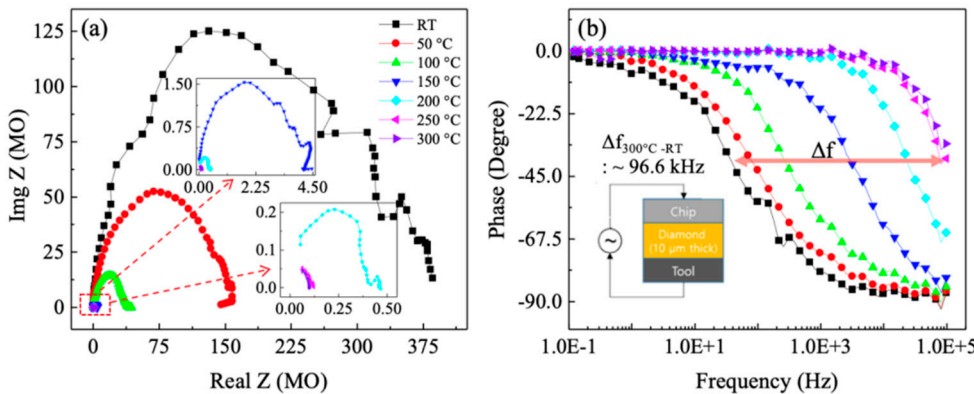

**Figure 2.** Impedance spectroscopy results under varying temperature conditions: (**a**) Nyquist plot and (**b**) phase diagram.

Capacitance under given temperature conditions can be estimated from the obtained Nyquist diagram where the capacitance is the inverse of the product of natural frequency and the real part of impedance. As shown in Table 1, although the estimated capacitance increased slightly from when at room temperature, the variance among given temperatures was minimal considering the capacitance effect. The capacitance was steady enough within the variation range of 5.8 pF under 300 °C. Therefore, capacitance was negligible and assumed to be constant in the equivalent circuit model; and only the real part of impedance, which is the leakage resistance as described in Figure 1, is considered for the study.

**Table 1.** Estimation of capacitance from Nyquist diagram.

| Temperature (°C) | RT | 50 | 100 | 150 | 200 | 250 | 300 |
|---|---|---|---|---|---|---|---|
| Capacitance (pF) | 8.8 | 10.2 | 13.4 | 7.7 | 11.1 | 13.5 | 14.5 |

Assuming capacitance is constant regardless of temperature variation, only the real part of impedance from the impedance spectroscopy is considered and compared with varying temperature conditions as shown in Figure 3. The result showed a non-linear exponential relationship between the impedance and temperature. Impedance of the dielectric layer exhibited temperature dependent characteristics, and the impedance was exponentially decreased as temperature rose. A non-linear relationship between temperature and impedance can be modeled by an exponential curve fitting. The unit of impedance on the Y axis is in mega ohm (MΩ). There should be a few tens of kilo ohm changes between 250 and 300 °C conditions. Although the experiment was conducted under 300 °C due to the limited laboratory setting, such differences are sufficient to differentiate the temperature conditions precisely. Thus, thermal characterization of the dielectric coating layer for the machining in-process cutting temperature monitoring method at the TCI region is tested throughout the experiment. In other words, the preliminary results showed that interface cutting temperature can be monitored based on impedance of the cutting tool.

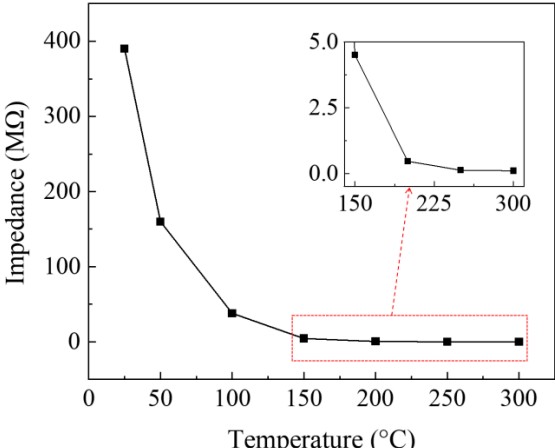

**Figure 3.** Impedance vs. temperature from Nyquist plot.

## 4. Experiments and Results

### 4.1. Voltage Controlled Current Source (VCCS)

A voltage-controlled current source (VCCS) was designed and implemented to monitor the impedance output with changes in temperature at the TCI. Due to the high impedance in the dielectric layer as previously seen from impedance spectroscopy results, a high impedance op-amp was used for building the VCCS. As shown in Figure 4a, the VCCS consists of the operational amplifier and P-channel MOSFET to ensure constant current flows through the system. The output voltage linearly increased as load resistance, $R_L$, increased—implying that the voltage is adequately controlled to provide constant current flow. The sensitivity with current setup used throughout the experiment was 64 mV/M$\Omega$ and the amplifier measured up to 150 M$\Omega$. The amount of constant current flow can be controlled by adjusting the input voltage ($V_{DC}$) at the op-amp as shown in Figure 4b. When the $R_L$ was fixed by controlling $V_{DC}$, impedance output could be adjusted, which itself indicates the amount of current was properly controlled. Depending on systems and machining conditions, the measuring range can also be changed by adjusting the shunt resistor, $R_S$, or $V_{DC}$ of the VCCS. From the experiment on the implemented VCCS, since the $R_L$ represents impedance at TCI, change in impedance at TCI could be monitored as $R_L$ changed, as output voltage measurement.

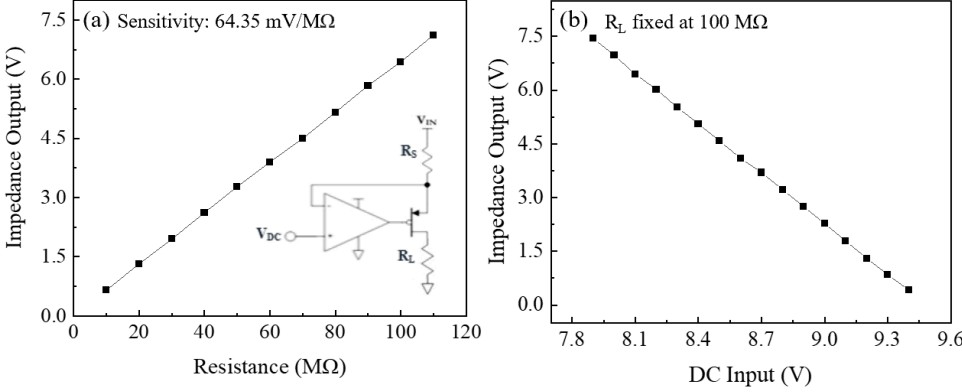

**Figure 4.** Voltage controlled current source (VCCS) testing result: (**a**) output voltage under varying $R_L$ when $V_{DC}$ is fixed, and (**b**) output voltage under varying $V_{DC}$ when $R_L$ is fixed.

### 4.2. Impedance vs. Temperature

Impedance of the dielectric layer under temperature variation was analyzed at the interface by placing the cutting tool in 50 μm contact with an aluminum (AL) workpiece while the AL was being gradually heated. A thermocouple (TC) was attached to the top

surface of the cutting tool near to where the contact was made for impedance comparison with temperature. The impedance response from VCCS and temperature from TC were measured simultaneously using a data acquisition device, and impedance was measured at $R_L$ in VCCS. Similar to the previous spectroscopy experiment result, TCI impedance showed an exponentially decreasing trend as temperature increased, as shown in Figure 5. Until temperature reached 40 °C, the impedance of 140 MΩ at RT dramatically decreased to 30 MΩ. After temperature exceeded 40 °C, changes in impedance showed an almost linear relationship with temperature variation. For repeatability, 3 trials of experiment were performed and all the results showed identical trends without showing any random response in impedance under the elevated temperature conditions. Therefore, from the thermal experiment, it is observed that the proposed impedance-based in-process cutting temperature monitoring method can be implemented into machining systems to monitor the TCI cutting temperature by measuring interface cutting temperature. This is achieved by directly probing the cutting tool in a workpiece.

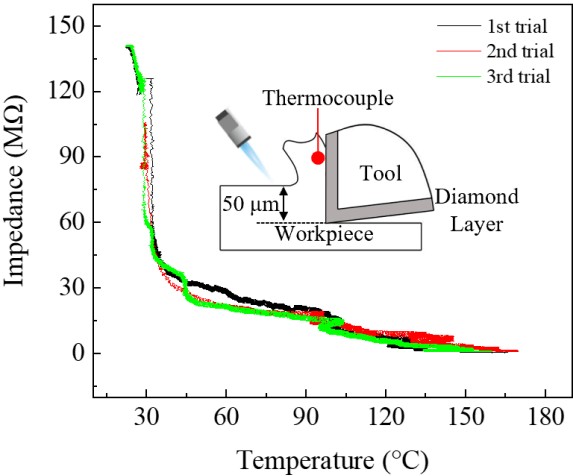

**Figure 5.** Thermal experiment.

### 4.3. Stability Test

To see the stability of impedance output, the experimental setup was constructed as shown in Figure 6. Contact depth of the cutting tool with AL was adjusted using manual stage; 5, 10, and 15 μm contact depths were tested while collecting impedance output from VCCS for 5 min. Outputs on all three cases were stable enough to monitor the cutting temperature and steady under corresponding contact depth. Additionally, no remarkable signal noise was detected other than some variations due to instable contact. Impedance was sensitive to not only temperature, but the contact area as shown in stability test results. The more contact made, the less impedance output was shown. Thus, it is expected that the proposed mechanism can also be utilized in the machining process monitoring method that precisely and accurately monitors vibration, chatters, or cutting force occurring in the interface.

### 4.4. Machining Experiment

The preliminary experiment on impedance of cutting tool with VCCS is followed by a machining experiment. In this experiment, the AL workpiece was turned on a lathe to sufficiently increase cutting temperature. Thus, AL material was cut by the cutting tool coated with a polycrystalline diamond layer, which could avoid wear effect. The identical cutting tool was used for the machining experiment as prior characterizations and constant current was supplied to the cutting tool by the VCCS. This is illustrated in Figure 7. With a feed rate of 0.1 mm/rev, three different rotational speeds (700, 800, and 900 rpm) were designated for the machining conditions. Since the horizontal tool speed varied with the spindle rotation speed to give the same feed rate, the material removal rate

(MRR), Q, was calculated for the evaluation of the interface cutting temperature under the nominal conditions. To electrically isolate the tool from the machine system, the polymer plate was inserted and securely fixed in between the tool and the system. The vertical tool traveling distance of the turning operations was around 60 mm, and the depth of the cut was set at 50 μm due to the limitation of the machining setup. The impedance output was acquired from the VCCS during the machining process. Figure 8 shows the machined surface after the turning operations observed by electronic microscope. However, no remarkable differences were observed. This became the primary reason for considering the direct probing in-process monitoring method.

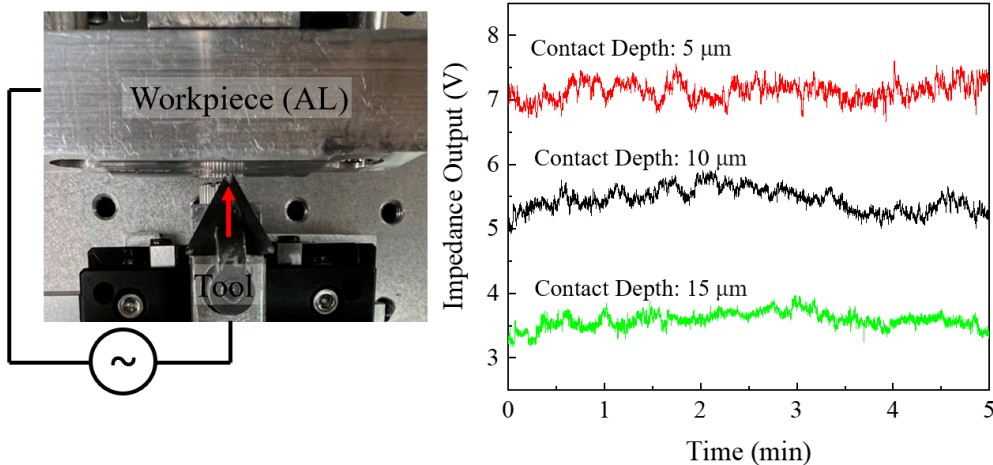

**Figure 6.** Experiment setup for stability test and results.

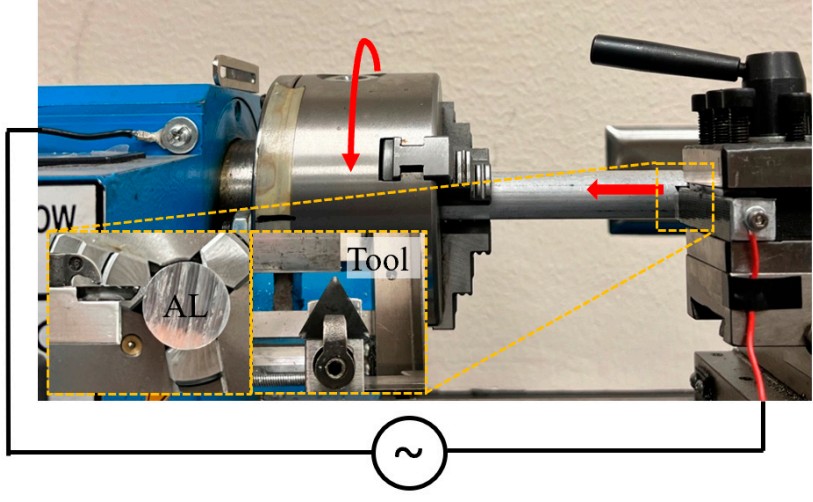

**Figure 7.** Turning machining experiment setup.

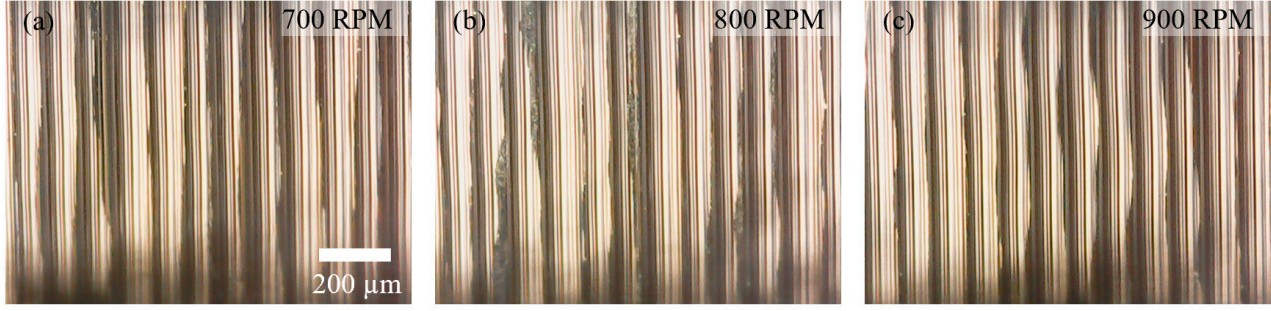

**Figure 8.** Microscope image of machined surface: (**a**) 700 rpm, (**b**) 800 rpm, and (**c**) 900 rpm.

Each machining condition showed different temperature profiles with the proposed in-process interface cutting temperature monitoring method, as shown in Figure 9. Impedance is inversely proportional to cutting temperature, which is predicted from the impedance spectroscopy result and thermal characterization. Thus, a drop in impedance implies a decrease in temperature. When machining with the lowest rpm condition corresponding to the material removal rate (MRR, denoted Q here) of 0.55 mm$^3$/min, impedance gradually decreased as more material was removed from the workpiece. However, when cutting the workpiece with the highest rpm condition that had MRR of 0.71 mm$^3$/min, the impedance rapidly dropped and saturated after cutting 30 mm. From the three given conditions, $\Delta T$ (which is the difference between the initial temperature and the temperature after machining 60 mm) was estimated to be 7.4, 9.6, and 75.7 °C at 700, 800, and 900 rpm, respectively. As more material is removed, the impedance decreases. In other words, cutting temperature increases when removing more material. Considering the difference in each of the machining conditions, the result showed that the proposed in-process machining monitoring method at TCI could monitor the cutting temperature and even capture minor differences in interface cutting temperature. Therefore, the method may be applied to conventional machine tools to monitor the machining process in a precise and accurate manner.

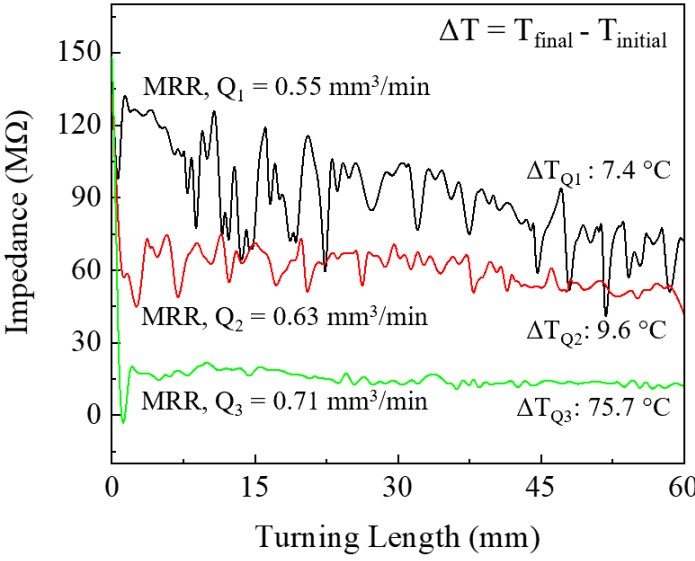

**Figure 9.** Impedance monitoring results during the turning process.

### 4.5. Potential Application for Indentation

The contact area of the cutting tool and workpiece during the machining process is another major factor in impedance response. As the contact area increases, the impedance of the dielectric coating layer decreases, as could be predicted from the stability test. The characterization of this effect was analyzed by measuring the impedance when the cutting tool is touching the AL workpiece. Similar to the stability test, the contact area was controlled by depth of the tool into the AL workpiece. It is evident from Figure 10 that impedance is also inversely proportional to the contact area at TCI. As the depth of contact increased, the impedance decreased from 7.8 to 2.8 G$\Omega$. This study is mainly focused on measuring the cutting temperature; however, the impedance is not only sensitive to temperature but also to the contact area. Therefore, it is expected that, with the proposed method, other machining processes such as chatter, vibration, and cutting force can be monitored while machining is in process. Additionally, the indentation process is one possible application that can be developed with the proposed method to measure material properties during the machining process. Also, the depth of the indentation process can be precisely estimated, and uniform patterns can be fabricated by monitoring the impedance feedback signal.

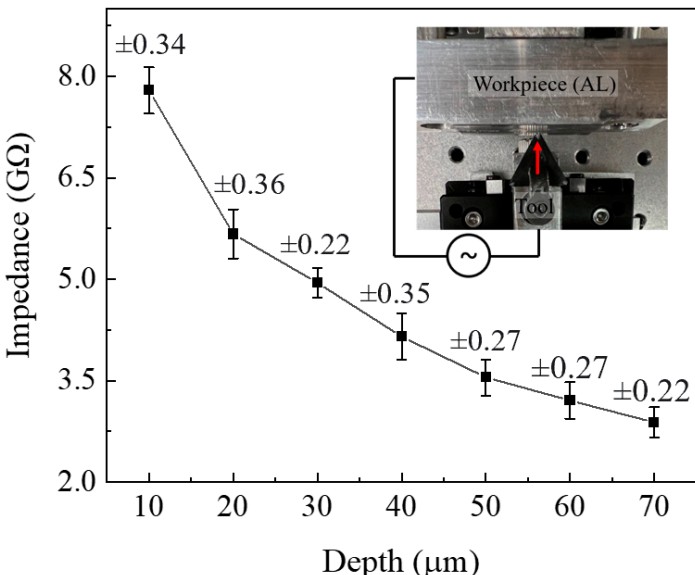

**Figure 10.** Impedance measurement under varying depth of contact.

In summary, the proposed method proved the feasibility and versatility of implementing an in-process machining impedance-based monitoring method. The machining process at the local TCI region is now expected to be directly monitored in a precise and accurate manner.

## 5. Future Works

Numerical and quantitative analysis for establishing an impedance model may convert the impedance into user-friendly output. Dynamometer and accelerometer data will be compared with impedance response during the machining process in our future study. Thus, identified impedance models could potentially assess optimal machining parameters and monitor various cutting mechanics that include cutting force, chip formation, tool vibration, and even micro/nanostructure changes [31–33]. Further experiments and applications will be conducted under different types of machining such as milling, facing, indentation, as well as machining with cooling methods, etc. including comparison between dry and wet machining conditions. Lastly, impedance characterization under different coating materials, shapes, and workpiece materials will be studied.

## 6. Conclusions

The impedance-based in-process machining monitoring method was introduced for the first time. It was concluded that the impedance of the dielectric coating layer was dependent on temperature as identified in the experiments of impedance characteristics. Hence, the proposed method proves that monitoring the cutting temperature at the local TCI is possible. Furthermore, the depth of contact between the cutting tool and workpiece is inversely proportional to the impedance change. The impedance change with respect to the contact area from the indentation result indicates the feasibility of vibration and chatter measurements occurring at the TCI. Additionally, prediction of chip formations and cutting force during the machining process can be identified. Thus, the study showed the feasibility of implementing the proposed method into conventional machining systems. In summary, the proposed impedance-based in-process machining monitoring method is expected to improve the control of cutting mechanics for advanced manufacturing by implementing it in dynamic systems to monitor the machining parameters and processes at the local TCI. The proposed method sheds light on analysis of cutting mechanisms and machine tool dynamics. In future work, the stainless steel and additively manufactured metals will be machined by using the precision machine tool while measuring the electrical impedance

at TCI. Then, the corresponding microstructure analysis of the machined parts as well as stress-strain changes of the cutting tool coating materials will be studied in-depth.

**Author Contributions:** H.C. conducted the design of experiment and circuit design. W.P. conducted the experimental setup, collected the data and analyzed the temperature-dependent impedance behavior. J.K. helped H.C. and W.P. conduct machining experiment, and analyzed the overall machining processes. The C.L. provided the concept of cutting temperature-impedance model and supervised the graduate students. All authors have read and agreed to the published version of the manuscript.

**Funding:** This research was supported by Walker Seed Grant (510502-00031) from J. Mike Walker '66 Department of Mechanical Engineering in Texas A&M University.

**Institutional Review Board Statement:** Not applicable.

**Informed Consent Statement:** Not applicable.

**Data Availability Statement:** Upon request, data will be provided.

**Acknowledgments:** This study was supported by MEEN Seed Grant by J. Mike Walker '66 Department of Mechanical Engineering in Texas A&M University and thanks to Choongho Yu and Jooyoung Lee for their aid throughout the experiments.

**Conflicts of Interest:** The authors declare no conflict of interest.

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
