# Peer review of "In-Process Cutting Temperature Monitoring Method Based on Impedance Model of Dielectric Coating Layer at Tool-Chip Interface"

_jmmp, doi:10.3390/jmmp6050097_

Round 1
Reviewer 1 Report
The article submitted for review is of high scientific importance, however, the authors need to clarify some practical issues:
1) The method of measuring the temperature in the contact zone of the tool and the workpiece, proposed by the authors, has a significant limitation associated with the nonlinear nature of the dependence of the Impedance on the cutting temperature. This is clearly seen from Figure 3, which shows the nonlinear saturation of the impedance dependence in the temperature zone above 150 degrees Celsius. At the same time, metal cutting is usually carried out in a higher temperature range. In addition to the saturation area, the characteristic area from 50 to 150 degrees is significantly non-linear, which will help lead to distortion of the measured signal. That is, the authors should indicate the limitations of the use of this temperature measurement method only by some options for finishing metals. There is also a question of applying this method to cutting processing options using cooling systems of the cutting zone.
2) The description of the experiment needs to be clarified, the material being cut is not specified here, as well as the cutting tool. It is also worth specifying the reason for choosing the nominal of this material and this cutting tool. From Figure 7 it is not possible to draw a conclusion about the methods of isolation of the measurement system (tool from the caliper) from possible electrical interference induced by the machine.
3) Figure 9 needs to be described additionally, here the authors are introducing the value Q, without explaining why it is named, and not the cutting speed characterizes the processing temperature. It follows from this figure that at a spindle rotation speed of 700 revolutions per minute, the increase in cutting temperature was only 7.4 degrees, and at a speed of 800 revolutions per minute of 9.6 degrees, the difference here was about 2 degrees. However, the following experiment gives an increase in temperature by 76 degrees, such an effect can only be explained by the measurement error. The authors of the article need to clarify the results obtained here.
4) The last 10 figure shows the dependence of the measured impedance value on the contact value of the tool and the workpiece. The authors believe that this is the advantage of the proposed measurement method, but in fact it is a disadvantage that shows the possible noise level of the measured temperature value by variations in the contact value of the instrument and the workpiece. That is, in the actual operating conditions of such a monitoring system, it will not be possible to determine the temperature variation or the contact area variation in the measured signal.
5) Taking into account the above comments, the authors need to deeply rework the conclusions of the work, indicating in them the advantages and disadvantages of their proposed method of measuring temperament.
Author Response
1) The method of measuring the temperature in the contact zone of the tool and the workpiece, proposed by the authors, has a significant limitation associated with the nonlinear nature of the dependence of the Impedance on the cutting temperature. This is clearly seen in Figure 3, which shows the nonlinear saturation of the impedance dependence in the temperature zone above 150 degrees Celsius. At the same time, metal cutting is usually carried out in a higher temperature range. In addition to the saturation area, the characteristic area from 50 to 150 degrees is significantly non-linear, which will help lead to distortion of the measured signal. That is, the authors should indicate the limitations of the use of this temperature measurement method only by some options for finishing metals. There is also a question of applying this method to cutting processing options using cooling systems of the cutting zone.
Answer: In many cases, the electrical properties of the metals show a highly non-linear relationship with temperature change. As the temperature increases, electrons from the valence band are able to jump to the conduction band, creating free movement between the two bands, thus the metal conductivity changes. Figure 3 shows the non-linear relationship between temperature and impedance, but this behavior can be modeled by an exponential curve fitting. The unit of impedance on Y axis is in mega Ohm (MΩ). There should be a few tens of kilo Ohm changes between 250- and 300 degrees Celsius conditions. Although the experiment was conducted under 300 degrees Celsius due to the limitation in the laboratory setting, such differences are sufficient to differentiate the temperature conditions precisely. The primary focus of this paper is to show the feasibility of a novel interface temperature measurement method. Thus, it is premature to set the limit of the proposed method. Further development and characterization of high temperature with the proposed method will be studied in future work. Additionally, we have preliminarily tested the cutting process under a cutting fluid condition and the result showed some impedance variations. However, it needs more study, and various cutting conditions under varying materials to be cut will be considered in future works. This paper introduces the fundamental method for measuring interface temperature that was not available. Thus, it is more focused on the testing of feasibility. The non-linear behavior and fitting methods were described in section 3.1 and the conclusion.
2) The description of the experiment needs to be clarified, the material being cut is not specified here, as well as the cutting tool. It is also worth specifying the reason for choosing the nominal of this material and this cutting tool. From Figure 7 it is not possible to draw a conclusion about the methods of isolation of the measurement system (tool from the caliper) from possible electrical interference induced by the machine.
Answer: Experiment details were rewritten for general readers including materials and cutting tool in sections 3.1 and 4.4. This study aims to test the feasibility of on-machine, in-process tool-chip impedance monitoring for cutting temperature monitoring. To avoid the cutting tool wear effect, the diamond-coated tool and aluminum material were chosen. Also, to isolate the tool from the machine system, a polymer plate was inserted and securely fixed in between the tool and the system. Descriptions are also added to the manuscript.
3) Figure 9 needs to be described additionally, here the authors are introducing the value Q, without explaining why it is named, and not the cutting speed characterizes the processing temperature. It follows from this figure that at a spindle rotation speed of 700 revolutions per minute, the increase in cutting temperature was only 7.4 degrees, and at a speed of 800 revolutions per minute of 9.6 degrees, the difference here was about 2 degrees. However, the following experiment gives an increase in temperature by 76 degrees, such an effect can only be explained by the measurement error. The authors of the article need to clarify the results obtained here.
Answer: Since the values of horizontal tool speed vary with the spindle rotation speed to have the same feedrate, material removal rate, Q, was calculated for the evaluation of temperature measurement under nominal values. As seen in Figure 5, tool-chip interface impedance was calibrated with respect to the given temperature at depth of cut 50 µm. The calibration results were highly repeatable. The result (Figure 9) shows that the cutting temperature increases as the material removal rate increases. In general, the cutting temperature changes dramatically as the cutting conditions change. It would be more important see the first few mm of turning length and changing in slope rather than looking final temperature changes. The material removal rate is the most dominant factor in determining the cutting temperature. This data was assessed based on the calibration result shown in Figure 5. Because there is no commercial reference-like cutting temperature measuring instrument, a direct comparison is not available. However, from the calibration result (Figure 5), the impedance value could indicate the current temperature in-process. Section 4.4 was updated.
4) The last 10 figure shows the dependence of the measured impedance value on the contact value of the tool and the workpiece. The authors believe that this is the advantage of the proposed measurement method, but in fact it is a disadvantage that shows the possible noise level of the measured temperature value by variations in the contact value of the instrument and the workpiece. That is, in the actual operating conditions of such a monitoring system, it will not be possible to determine the temperature variation or the contact area variation in the measured signal.
Answer: Figure 10 shows the impedance result on metal indentation. In general, indentation does not involve temperature. Figure 10 shows the impedance change according to the indentation depth, so this result indicates that tool-chip interface impedance monitoring enables the measurement of the indentation depth and characterizing indentation dynamics. Bi-material contact mechanism is very complicated, especially for dynamic conditions. Here authors wanted to show other applications of the proposed idea, tool-chip interface monitoring, for the indentation. Please note this result does not involve the temperature effect. In addition, even in the machining process, the contact mechanism cannot be seen as a disadvantage as the machining process will include chip vibrations at the interface which is unknown and need more study in the near future. Chip vibration will cause a change in contact area which can be thought of as noise, but in fact, it is machining dynamics. With changes in the contact area, the temperature can be sufficiently identified as already seen in figure 9. Section 4.5 was updated and was clearly described to avoid any potential confusion.
5) Taking into account the above comments, the authors need to deeply rework the conclusions of the work, indicating to them the advantages and disadvantages of their proposed method of measuring temperature
As the reviewer suggested, the conclusion part was rewritten, and the potential advantages and limitations were included.
Reviewer 2 Report
The article entitled "In-Process Cutting Temperature Monitoring Method Based on Impedance Model of Dielectric Coating Layer at Tool-Chip Interface" introduces a novel approach to in-process monitor the cutting temperature at the tool-chip interface (TCI). During the aluminium alloy turning process, the dielectric coating layer of the cutting tool is sandwiched in between the base material of the cutting tool and workpiece. The base material of the cutting tool and workpiece conduct electricity function as electrodes and the dielectric coating layer behaves as a dielectric capacitor.
In the authors' article, it would be necessary to describe in more detail the method of applying the mentioned layer to the interchangeable cutting inserts used in the turning process. As the presented experiment deals with a layer based on polycrystalline diamond, and the abstract also mentions other applied layers, e.g. based on CBN. In the methodology and experimental detail in the mentioned article, the method of applying such a layer is not indicated, it is not completely clear whether you can use commonly available cutting plates available on the market for such a process, or it is necessary to prepare special cutting inserts. At the same time, it would be appropriate for the readers of the article to publish the method in which the mentioned layer is applied, in what way it is applied, there is no publication of the inserts with the original PVD or CVD layers applied with a new layer, through the required microscopy methods. A comparative study of steel turning could also be presented at the same time, but this is not possible with a polycrystalline diamond layer due to affinity. The authors of the article also investigated the strengthening of steels using the CBN dielectric layer, as they write and present in the abstract of the article.
Author Response
In the authors' article, it would be necessary to describe in more detail the method of applying the mentioned layer to the interchangeable cutting inserts used in the turning process. As the presented experiment deals with a layer based on polycrystalline diamond, and the abstract also mentions other applied layers, e.g. based on CBN.
Answer: We used a commercially available chemically deposited diamond coating layer. In order to avoid tool wear during the experiment, the diamond coating was considered. Tool-chip interface impedance can be measured on-machine and in-process during the machining. The tool-chip interface needs to be electrical dielectric. The main idea of this study is that the electrical property of the tool-chip interface (dielectric layer) changes according to the given temperature so that the cutting temperature can be on-machine, and in-process measured. In the Abstract, the tool coating information was added and revised to avoid any confusion.
In the methodology and experimental detail in the mentioned article, the method of applying such a layer is not indicated, it is not completely clear whether you can use commonly available cutting plates available on the market for such a process, or it is necessary to prepare special cutting inserts.
Answer: As already motioned, the commercially available cutting tools were purchased. Diamond-coated tools are wear-resistive and highly dielectric, even under high temperatures. Also, such tools work for dry-condition machining. Tool information (manufacturer, model, material, and layer thickness) was added in section 3.1.
At the same time, it would be appropriate for the readers of the article to publish the method in which the mentioned layer is applied, in what way it is applied, there is no publication of the inserts with the original PVD or CVD layers applied with a new layer, through the required microscopy methods.
Answer: As we purchased a commercially available tool, we have not done any post-processing to the cutting tool and the tool was used without any modification from the manufacturer. The manufacturer only gave us the information of whether it is PVD, or CVD coated with a rough estimation of coating depth. Information on cutting tools is provided in section 3.1 and the abstract.
A comparative study of steel turning could also be presented at the same time, but this is not possible with a polycrystalline diamond layer due to affinity. The authors of the article also investigated the strengthening of steels using the CBN dielectric layer, as they write and present in the abstract of the article.
Answer: This paper represents tool-chip interface impedance monitoring for assessing cutting temperature on-machine and in-process. Due to the limited machining environment, only aluminum materials were cut. Because the used cutting tools have a diamond coating, cutting the steel is not recommended. In future work, the various dielectric layers such as ceramics or CBN will be considered to cut hard materials such as steel, stainless steel, or titanium. Future work was included in the section of Conclusion.
Round 2
Reviewer 2 Report
The submitted corrected article by the authors was edited globally according to the reviewer's instructions. The authors explained the reason for choosing the al alloy for the experiment and subsequently the type of cutting insert used. However, the article could be supplemented with at least the microstructure of the machined Al alloy, or microstructural analysis of the cutting insert, mainly focusing on its coating. In the same way, the list of literary sources would like, but does not have to be expanded by a higher number of sources in the final result.
Author Response
Thank you for your good comments on this manuscript.
We revised this manuscript and went through extensive English revisions by professional English editing software. Below is the answer to the question raised in this review process.
Question: The submitted corrected article by the authors was edited globally according to the reviewer's instructions. The authors explained the reason for choosing the al alloy for the experiment and subsequently the type of cutting insert used. However, the article could be supplemented with at least the microstructure of the machined Al alloy, or microstructural analysis of the cutting insert, mainly focusing on its coating. In the same way, the list of literary sources would like, but does not have to be expanded by a higher number of sources in the final result.
Answers: To the best of our knowledge, a too-chip interface (TCI) impedance monitoring technique was first introduced in this study. This study is mainly focused on impedance-temperature calibration and preliminary tests in a lab-scale machining environment. The machine tool used in this study is a small lathe, the so-called mini-lathe, which only works for wood or aluminum cutting. Thus, the surface quality of the machined part is not as smooth as the precision machine tool cuts. In future work, the stainless steel and additively manufactured metals will be machined by using the precision machine tool while measuring the electrical impedance at TCI. Then, the corresponding microstructure analysis of the machined parts and even stress-strain changes of the cutting tool coating materials will be studied in-depth.
This future work was briefly addressed in the section of the Conclusion.